# Deep-Fried Atractylodis Rhizoma Protects against Spleen Deficiency-Induced Diarrhea through Regulating Intestinal Inflammatory Response and Gut Microbiota

**DOI:** 10.3390/ijms21010124

**Published:** 2019-12-23

**Authors:** Kun Shi, Linghang Qu, Xiong Lin, Ying Xie, Jiyuan Tu, Xianqiong Liu, Zhongshi Zhou, Guosheng Cao, Shuiqing Li, Yanju Liu

**Affiliations:** 1College of Pharmacy, Hubei University of Chinese Medicine, Wuhan 430065, China; s1062033640@163.com (K.S.); qulinghang@163.com (L.Q.); 15672693563@163.com (X.L.); 18751028560@163.com (Y.X.); 3070@hbtcm.edu.cn (J.T.); 1424@hbtcm.edu.cn (X.L.); moneyzzs@163.com (Z.Z.); caoguosheng2006@163.com (G.C.); mississippiwithme@163.com (S.L.); 2Center for Hubei TCM Processing Technology Engineering, Wuhan 430065, China

**Keywords:** diarrhea, gut microbiota, P38, MAPK signaling pathways

## Abstract

According to the theories of traditional Chinese medicine, spleen deficiency often leads to diarrhea, and deep-fried Atractylodis Rhizoma (DAR) is commonly used for the treatment. However, the association between spleen deficiency and diarrhea remains unclear. The present study aimed to investigate the therapeutic effect of DAR for the treatment of diarrhea caused by spleen deficiency and analyze the related mechanisms. It was found that a high dose group of an ethanolic extract of deep-fried Atractylodis Rhizoma (EEDAR-H) significantly inhibited weight loss, diarrhea, and pathological changes in colon tissue induced by rhubarb. EEDAR-H was found to significantly reduce the level of intestinal inflammatory cytokines and increase the expression of gastrointestinal motility hormones. In addition, EEDAR-H significantly increased the expression of aquaporin 3 (AQP3) and aquaporin 8 (AQP8) and restored abnormal water metabolism; Shen-Ling-Bai-Zhu-San (SLBZS) induced the same effect as EEDAR-H. Additional tests on the mechanism found that EEDAR-H and SLBZS promoted the integrity of the intestinal barrier. Both significantly increased the expression of the tight junction protein ZO-1 and Occludin, inhibited the phosphorylation of p38MAPK and MLC, and significantly reduced the expression levels of PAR-2. Analysis of the gut microbiota indicated that overall changes in its structure were reversed after treatment with EEDAR-H or SLBZS, in addition to significant modulation of the abundance of different phyla. At the genus level, EEDAR-H or SLBZS significantly reduced the levels of potential pathogens and increased those of beneficial bacteria.

## 1. Introduction

Spleen deficiency-induced diarrhea syndrome is a common sub-health status described in traditional Chinese medicine (TCM). The main cause of spleen deficiency diarrhea according to TCM is considered to be Qi consumption, spleen deficiency, stomach disharmony, and endophytic dampness, resulting in conductive dysfunction of the intestines [1]. The treatment of spleen deficiency diarrhea is based on the strategy of supplementing Qi and strengthening the spleen. In TCM, the clinical manifestation of spleen deficiency-induced diarrhea syndrome consists of diarrhea (main symptom), weight loss, chills, and fatigue, which are secondary manifestations of multiple symptoms. The syndrome also includes a variety of symptoms similar to those found in inflammatory bowel disease, chronic diarrhea, and ulcerative colitis. In China, Shen-Ling-Bai-Zhu-San (SLBZS) is widely used to treat diarrhea caused by spleen deficiency. A number of studies have shown that SLBZS can alleviate diarrhea caused by spleen deficiency in various ways, such as targeting inflammation and flora [2,3,4].

Modern medical research has shown that the occurrence and development of spleen diarrhea is closely related to gastrointestinal digestive dysfunction, intestinal inflammatory lesions, low immune status [5,6], and disorders of water metabolism [7,8]. The organ principally involved in diarrhea is the intestine. To function normally, it requires a correctly functioning intestinal barrier, an appropriate quantity of microbiota, and a controlled level of inflammation. These 3 components of intestinal physiology are interrelated: increased intestinal permeability can lead to a rise in the level of endotoxins or the transport of endotoxins (produced by the microbiota) into the circulatory system. This in turn triggers systemic endotoxin build up and produces an inflammatory response [9,10,11]. Microbial changes can affect intestinal barrier function, endotoxin production, and synthesis of nutrients and energy-regulating hormones. Restoration or maintenance of the integrity of the intestinal barrier and microbial balance may be effective strategies for optimizing metabolism and preventing the development of pathologies associated with metabolic disorders [12,13,14]. Studies have shown that the integrity of the intestinal barrier is mainly related to the normal activity of tight junction proteins. Previous studies have shown that the expression of ZO-1 and Occludin is significantly reduced in diarrheal diseases [15,16]. In addition, it has been found that PAR-2 and p38MAPK are involved in the regulation of tight junction protein expression [17]. PAR-2 can promote the phosphorylation of p38 after activation, which in turn can promote the activation of myosin light chain kinase and myosin light chain, which then causes the translocation or degradation of tight junction proteins [18,19,20]. Healthy gut microbiota contributes to intestinal mucosal protection, metabolism, immune homeostasis and response, and pathogen suppression [21,22]. Previous studies have shown that the intestinal flora in patients with diarrhea changes significantly, with the abundance of beneficial bacteria such as Lactobacillus and Akkermansia decreasing significantly, while harmful bacteria such as Clostridium and Helicobacter become significantly more abundant. Diarrhea can effectively be alleviated by inhibition of the growth of the harmful microflora [23,24].

Deep-fried Atractylodis Rhizoma (DAR), a traditional Chinese medication, strengthens the spleen and has anti-diarrheal properties. It is often used to treat spleen deficiency-induced diarrhea. Unlike traditional Western medicine, Chinese herbal medicines are commonly used to treat patients in a holistic manner. As systems biology explores the complex interactions between the components of biological logic systems, the study of specific symptoms of TCM and the mechanisms of action of herbs have significant advantages [25]. Little is known about the effects and the molecular mechanism of DAR due to the diversity of the ingredients of medications used in TCM and the complexity of the interaction between drugs and the human body. In addition, Chinese herbal medicine differs from modern Western medicine in terms of substance, method, and philosophy, and hinders Western countries from recognizing and accepting the therapeutic benefits of herbal medicine. Therefore, we urgently need to understand the therapeutic effects and mechanisms of action of DAR on spleen deficiency diarrhea syndrome and provide a basis for the rational application of DAR.

## 2. Results

### 2.1. EEDAR Inhibited Changes in Body Weight, Diarrhea Score, and Fecal Water Content in Spleen Deficiency Diarrhea Rats

As shown in (Figure 1A), the weights of rats in the experimental models of diarrhea and diarrhea with rhubarb decreased significantly. After treatment with EEDAR-H or SLBZS, their weights increased, values that were significantly different from those in the model group (*p* < 0.01). Moreover, compared with the model group, EEDAR-H and SLBZS significantly reduced the diarrhea grade score (*p* < 0.01) (Figure 1B) and fecal moisture content (*p* < 0.01) (Figure 1C). However, these effects were not observed after treatment with EEDAR-L.

### 2.2. EEDAR Inhibited Pathological Changes and Inflammatory Cytokines in the Colons of Spleen Deficiency Diarrhea Rats

As illustrated in (Figure 2A,B), the level of inflammation in rats of the model group increased significantly. In comparison, EEDAR-H and SLBZS significantly inhibited accumulation of these inflammatory factors and also promoted the expression of the anti-inflammatory factor IL-10 (*p* < 0.01). From hematoxylin-eosin stained sections, it can be seen that compared with the colons of rats in the model group, pathological changes (inflammatory cell infiltration and intestinal structural abnormalities) were not found in the EEDAR-H and SLBZS groups. The findings were similar to those observed in the control group. Although there was no apparent damage to the intestinal structure of EEDAR-L, there was still a large degree of inflammatory cell infiltration.

### 2.3. EEDAR Increases the Levels of Gastrointestinal Motility Hormones and Inhibits the Expression of Serotonin and Its Receptors

As shown in (Figure 3A,B), the levels of gastrointestinal motility hormones were low due to weak spleen and stomach function in the model group (*p* < 0.05). After treatment with EEDAR-H or SLBZS, the levels of MTL and GAS returned to normal, and gastrointestinal function improved, with significant differences compared with the model group. In addition, serotonin (5-HT) and its receptors were measured, finding that in the model group, 5-HT levels increased significantly (*p* < 0.05) and the expression of its receptor also increased significantly. After treatment, both decreased significantly in the EEDAR-H and SLBZS groups (Figure 3C,D). In the EEDAR-L group, although a decrease was observed, the difference was not significant compared with the model group.

### 2.4. EEDAR Reduces the Expression of Aquaporin3 (AQP3) and Aquaporin8 (AQP8)

Diarrhea is usually accompanied by abnormalities in water metabolism in the intestine. Therefore, AQP3 and AQP8 are expressed in large quantities in the intestine and control the influx of water. We examined AQP3 and AQP8 in colon tissue and found that the expression of AQP3 and AQP8 decreased significantly in the model group (*p* < 0.05). In comparison, AQP3 and AQP8 increased significantly in the EEDAR-H and SLBZS groups (*p* < 0.01) (Figure 4A,B) and were higher even than the control group. Similar results were observed in the immunohistochemical sections (Figure 4C).

### 2.5. EEDAR Prevents Changes in TJ Proteins in the Colons of Rats with Diarrhea Due to Spleen Deficiency

Results of the previous experiments demonstrated that spleen-deficiency diarrhea rats have intestinal inflammation and intestinal damage and that such damage is usually related to the intestinal barrier [9]. Therefore, we analyzed colon tissue to detect the expression of the tight junction proteins, ZO-1 and Occludin. The results indicated that the expression of ZO-1 and Occludin decreased significantly in the model group. In comparison, following treatment with EEDAR-H or SLBZS, the expression of ZO-1 and Occludin increased significantly (Figure 5A,B). Although their levels tended to increase in the EEDAR-L group, there was no significant difference in comparison with the model group.

### 2.6. EEDAR Modulates Signaling Pathways That Affect TJ Structure and Function

Previous studies have shown that both the P38 and PAR-2 pathways are involved in the modulation of TJ structure and function [26]. To evaluate the activation of these cascades, we measured the phosphorylation of P38 and the expression of PAR-2 by Western blot analysis. P38 phosphorylation and PAR-2 levels increased significantly in the colon of spleen-deficiency diarrhea rats compared to those observed in controls (Figure 6A,B). Similar results were observed by immunohistochemistry (Figure 6C). In addition, we also tested myosin light chain kinase (MLCK) and phosphorylated myosin light chain (P-MLC) levels. The results indicated that MLCK and P-MLC increased significantly in the model group. After treatment with EEDAR or SLBZS, both P-MLC and MLCK decreased. Interestingly, EEDAR-L and EEDAR-H caused the same effect on MLCK and P-MLC (Figure 6E,F).

### 2.7. EEDAR Modulates the Overall Structure of Gut Microbiota in Spleen-Deficiency Diarrhea Rats

In order to detect differences in intestinal microbial structure between the 5 groups, a total of 863,427 sequences representing a mean of 57,561 sequences per group were obtained from fecal samples using an Illumina HiSeq sequencing system. (Figure 7A) shows that both the number of samples and the abundance of species were stable in a species accumulation box plot. Sequences were clustered into OTUs at a similarity of 97%, with a total of 882 OTUs obtained from 15 fecal samples. The overall evaluation of colon microbiota profiles was accomplished by principal components analysis (PCA), which indicated that the cluster representing the microbiota of the model group animals were significantly separate from those of the control rats, while the spleen-deficiency diarrhea rats supplemented with EEDAR-H or SLBZS exhibited clustering similar to that of controls. Interestingly, fecal microbiota from rats treated with EEDAR-L demonstrated clear cluster separation in comparison with controls (Figure 7B).

In addition, the compositional similarity of intestinal microbes for each group was analyzed using a cluster heat map. The results demonstrate that the species richness of the EEDAR-H and SLBZS groups was significantly greater than that of the model group and similar to that of the control group. The effects of treatment with EEDAR-H and SLBZS on the abundance of intestinal flora in spleen-deficiency diarrhea rats were similar. There was a high level of similarity between the EEDAR-H, SLBZS and control groups but with different species diversity (Figure 7C). In addition, a statistical analysis was conducted on the principal phylum and genus of each group of microorganisms. The results revealed that the abundance of Firmicutes and Proteobacteria changed significantly in the model group. In addition, there was a significant increase in various genera such as Clostridium, Bacteroides, Parabacteroides, Alloprevotella, and Helicobacter rats of the model group, increasing the probability of the development of diarrhea. Following treatment with EEDAR-H or SLBZS, the number of Lactobacillus and Akkermansia increased significantly compared with the model group (Figure 7D,E).

## 3. Discussion

Spleen deficiency diarrhea syndrome is a disease commonly seen by practitioners of TCM, especially in people with irregular diet, work, and rest [27]. The application of DAR for the treatment of diarrhea due to spleen deficiency started in the Song Dynasty, but with limited development due to the lack of defined pharmacodynamics and mechanisms. In the present study, a model of spleen deficiency diarrhea was selected that matched the characteristics of TCM syndrome to explore anti-diarrhea efficacy and mechanisms of alcohol extract of DAR.

MTL and GAS are two important hormones that affect gastrointestinal digestion. GAS can stimulate the secretion of gastric acid and pepsinogen [28,29]. MTL mainly acts on gastrointestinal smooth muscle and promotes the function of gastrointestinal strength. Force contraction and small bowel movements have a direct influence on the rate of gastric emptying and the small intestine propulsion rate [30]. Studies have shown that spleen deficiency leads to decreased expression of GAS and MTL [31]. Therefore, MTL and GAS levels were chosen as objective indicators for the efficacy of “enhanced spleen.” In the present study, we found that EEDAR-H significantly increased the levels of MTL and GAS and improved gastrointestinal function, consistent with data from previous studies. 5-HT is widely distributed in the central nervous system and gastrointestinal tract. It is involved in the regulation of gastrointestinal motility, secretion, sensation, and other functions. It mainly mediates various stress-related changes such as stress-induced high colonic motility, water-like diarrhea, and sensitivity to colonic dilatation [32]. A subtype of the 5-HT receptor, 5-HT3R can mediate the rapid activation of sensory afferent fibers. It is closely related to the occurrence of abdominal pain and diarrhea [33]; 5-HT3R can cause visceral pain stimulation resulting in abdominal pain, contraction of the ileum and colon, enhancement of gastrointestinal motility, and secretion, thereby causing an increase in fecal frequency and fecal water-like appearance [34,35]. The results indicate that EEDAR-H can simultaneously inhibit the expression of 5-HT and its receptors, thus alleviating the symptoms of diarrhea and abdominal pain, suggesting that the mechanism of EEDAR may be related to the reduced secretion of 5-HT.

Aquaporin is a protein located on the membranes of cells that controls the ingress and egress of water [36]. Studies have demonstrated that AQP3 and AQP8 can promote the reabsorption of excess water in the intestine, reduce the water content of feces, and control diarrhea [37,38]. In the present study, we found that EEDAR-H significantly promoted the expression of AQP3 and AQP8 and so relieved diarrhea. In addition, we also found that levels of the EEDAR-H and SLBZS groups were significantly higher than those in the control group, possibly related to the promotion of AQP expression by EEDAR under normal conditions.

Diarrhea is usually accompanied by intestinal damage, principally related to the intestinal barrier [39]. Previous studies have shown that its integrity is mainly determined by tight junction proteins [40,41]. Here, we found that EEDAR-H promoted the expression of ZO-1 and Occludin and protected the integrity of the intestinal barrier. As the p38 MAPK signaling pathway is involved in regulating the expression of tight junction proteins, and PAR-2 can promote the phosphorylation of P38 [42,43], we measured the expression of both proteins by immunoblotting. The results demonstrated that EEDAR-H reduced the phosphorylation of p38 by inhibiting the expression of PAR-2. MLCK can promote the phosphorylation of MLC, induce contraction of actin and myosin, promote the endocytosis of Occludin and other proteins, and then cause the translocation and reduction of tight junction proteins [44,45,46]. The results of the present study demonstrated that EEDAR inhibited both P-MLC and MLCK.

Recently, the gut microbiota has been recognized to be critical in influencing the in vivo therapeutic effects of herbal medicines [47,48]. Gut microbiota analysis with 16S rDNA amplification sequencing revealed significant differences in the overall structure of the gut microbiota between different experimental groups, especially in rats of the model group compared with those in the control group. PCA demonstrated that the EEDAR-H and SLBZS groups possessed an intestinal micro-ecological environment similar to that of the control group. However, those in the model and control groups were considerably different. At the phylum level, Firmicutes, Bacteroidetes, and Proteobacteria were the 3 dominant bacteria found in the feces of control rats in the present study. Bacteroidetes are considered to be mucosal-associated microbiota, and Firmicutes and Proteobacteria are regarded as fecal-associated microbiota [49,50]. Abundant variations of these are able to render intestinal mucosa more sensitive to bacterial infection and trigger gastroenteric diseases such as diarrhea and enteritis [51]. At the genus level, pathogenic bacteria such as Helicobacter pylori and Clostridium increased significantly but decreased significantly after treatment with EEDAR-H, and beneficial bacteria such as Lactobacillus and Akkermansia increased significantly, indicating that EEDAR-H can promote the growth of beneficial bacteria and restore intestinal micro-ecology, suggesting that this represents a novel treatment for diarrhea.

## 4. Materials and Methods

### 4.1. Extraction of Rhubarb and DAR

Atractylodis Rhizoma and rhubarb were obtained from the Hubei Tianji Chinese Herbal Pieces Company (Wuhan, China). The authenticity of the medicinal materials was confirmed by Xiu Qiao Zhang, an associate Professor from the College of Pharmacy of Hubei University of Traditional Chinese Medicine. DAR was prepared in accordance with instructions from the 2009 edition of Hubei Province processing specifications. Atractylodis Rhizoma (150 g) was heated to a temperature of 220–230 °C, stir-fried for 6 min at a rate of 50 times/min. Fifty g of the resultant DAR were then crushed through a No. 2 sieve, soaked overnight with 10 fold its volume in 80% ethanol, then extracted 3 times in an ultrasonic water bath, each for 2 h, filtered, and the extracts combined. The alcohol was recovered, and the DAR condensed to 740.4 mg/mL for use with the high-dose group, then partially diluted to 185.1 mg/mL for use as a low-dose group. The rhubarb was chopped, and an appropriate volume of water added. The mixture was soaked in an oven at 60–80 °C overnight, and then squeezed to obtain juice, filtered using gauze, then concentrated by steaming to prepare a liquid solution with a concentration of 2 g/mL. Original vouchers for the Atractylodis Rhizoma and rhubarb were deposited in the Hubei University of Traditional Chinese Medicine (Voucher nos. Atractylodis Rhizoma: 20180712; Rhubarb: 20181107).

### 4.2. In Vivo Experimental Design

Specific Pathogen Free (SPF) graded Wistar rats (140 ± 20 g) were provided by the Experimental Animal Research Center of Hubei Province (SYXK (e) 2017-0067). The animals were housed within a standard environment at a temperature of 25 ± 2 °C, humidity of 55 ± 5%, and a light/dark cycle of 12 h/12 h. The rats were provided with a standard diet and water. They were acclimatized to the new conditions for 5 days prior to the experiment. All 50 rats were randomly divided into 5 groups: control, model, EEDAR-L, EEDAR-H, and SLBZS. The appropriate quantity of SLBZS was added to water to prepare a suspension with a concentration of 246.8 mg/mL, representing the positive control. Except for the control group, all other groups were used as a treatment in the spleen-deficiency diarrhea model, with rhubarb-derived liquid administered to the stomach at a dose of 20 mL/kg, twice per day, morning and evening at an intervals of 8 h, for a total of 10 days. Each group was then administered the appropriate treatment at a dose of 10 mL/kg, daily for 7 days. The whole experiment’s duration was 17 days. During the experiment, the same volume of normal saline was administered to the control group. On the last day of the experiment, fecal samples were collected from the rats and stored at −80 °C. The experimental design was in strict accordance with the principles and guidelines recommended by the Chinese Association for Laboratory Animal Sciences (CALAS) and approved by the Animal Ethics Committee of Hubei University of Traditional Chinese Medicine (approval: NO.00273280, 10 November 2018).

### 4.3. Record of Body Weights, Diarrhea Scores, and Fecal Moisture Content

During the experiment, body weights, diarrhea scores and the rats’ fecal moisture content were measured at fixed times and locations every day. The evaluation of the diarrhea scores was based on the method of Akinobu Kurita [52]: 0, no diarrhea or normal stools; 1, mild diarrhea with wet and soft stools; 2, moderate diarrhea with dilute stools and mild perianal stains; 3, severe diarrhea with watery stools and severe perianal stains.
Fecal moisture content = (wet quality before drying − dry mass after drying)/wet mass before drying × 100%.

### 4.4. Observation of Intestinal Pathology

Intestinal segments of the colons of rats, approximately 4 cm in length, were harvested and fixed in 4% formaldehyde, embedded in paraffin, sliced into 4 µm-thick sections and stained with hematoxylin-eosin (HE). For IHC, colon tissues were fixed in a 4% (*w*/*v*) solution of paraformaldehyde in PBS overnight, rinsed with PBS, then stored in 70% (*v*/*v*) ethanol. Samples were embedded in paraffin, and 5 μm-thick sections created. Once deparaffinized, the sections were processed for antigen retrieval by placing in 10 mM sodium citrate buffer (pH 6.0) containing 0.05% (*v*/*v*) Tween 20 at 95 °C for 10 min, washed twice in 0.1% (*v*/*v*) Triton X-100 in PBS, blocked for 45 min in 2% (*v*/*v*) donkey serum in 0.1% (*v*/*v*) Triton X-100 in M PBS, then incubated overnight at 4 °C with primary antibodies for 5-HTR3, AQP8, P-p38, AQP3, or PAR-2 (1:200). Sections were washed in PBS, then incubated for 2 h at room temperature with Cy3-conjugated donkey anti-mouse or anti-rabbit IgG antibody (1:500). After immune-staining, cell nuclei were stained with 1 μg/mL Hoechst 33342; then, sections were imaged using an Olympus FV 1000 laser scanning confocal microscope (Olympus, Tokyo, Japan). Olympus FluoView version 4.0 software (Olympus, Inc., USA) was used to merge images.

### 4.5. Western Blot Analysis

Western blot analysis was conducted using standard protocols. After quantitative analysis, equal quantities of proteins were loaded onto and separated using sodium dodecyl sulfate-polyacrylamide gel electrophoresis (SDS-PAGE) on 8% to 12% gels, then transferred to a 0.45 μm pore-sized polyvinylidene fluoride (PVDF) membrane (Hercules, CA, USA). After blocking with 5% bovine serum albumin (St. Louis, MO, USA), membranes were incubated with primary antibodies (1:500) (Danvers, MA, USA) overnight at 4 °C, washed with TBS-Tween 20 buffer, then incubated in secondary antibody (1:2000) (Cambridge, MA, USA) for 2 h at 4 °C. Positive bands were visualized in accordance with a previously published method [53] using an enhanced chemiluminescence (ECL) detection system (Carlsbad, CA, USA).

### 4.6. Enzyme-Linked Immunosorbent Assays (ELISA)

After the final administration of each treatment, rats were fasted for 12 h, and their abdominal aorta anesthetized with 10% chloral hydrate. Samples were centrifuged at 3000 r/min for 20 min. Serum was separated from clotted blood. In addition, colon tissue was measured for AQP3 and AQP8. The concentrations of TNF-α, IL-10, MTL, 5-HT, and GAS in serum were quantified. All ELISA kits were purchased from Elabscience Biotechnology Co. Ltd. (Wuhan, China).

### 4.7. 16S rDNA Gene Sequencing and Analysis

Firstly, genomic DNA from fecal samples was extracted using sodium dodecyl sulfate (SDS). The concentration of DNA was then measured using agarose gel electrophoresis. The 16S rDNA V3-V4 region was selected as the amplification region. Universal primers were used and an index and linker sequences suitable for MiSeq PE300 sequencing were added at the 5′ end of the universal primer to complete the design of the specific fusion primer. A mixed library was subjected to gelatinization purification using a QIA-quick gel recovery kit. After recovery, the library was examined and quantified using a Fragment Analyzer with an Applied Biosystems Quant Studio 6 real-time PCR instrument. Finally, the library was processed on an Illumina HiSeq 2500 platform. After performing QC on the original data, U search software was used to de-chimerize and cluster the data. Each operational taxonomic units (OTU) was considered a distinct species. The reads of each sample were then randomly leveled and the corresponding OTU sequence extracted. After categorization, according to the number of sequences in each OTU, an OTU abundance table was created, and finally, analysis performed according to this OTU abundance table.

### 4.8. Other Materials

Primary antibodies for β-actin (#12620), phosphor (Thr180/Tyr182) p38 (#4511), p38 (#9212), phosphor (Thr18/Ser19) MLC (#3671), MLC (#3672), and PAR-2(#6976) were obtained from Cell Signaling Technology (Danvers, MA, USA). Antibodies for ZO-1 (ab216880), Occludin (ab167161), MLCK (EP1458Y), AQP3 (ab153694), secondary antibodies including anti-rabbit (ab6721), and anti-mouse (ab6728) IgG antibodies conjugated to horseradish peroxidase were obtained from Abcam Inc (Cambridge, MA, USA). Antibodies for 5-HT3R and AQP8 were purchased from Bioss Inc (Beijing, China). Enhanced chemiluminescence (ECL) Western blotting system was acquired from Thermo Fisher Scientific Inc (Piscataway, NJ, USA). 

### 4.9. Statistical Analysis

The results are presented as means ± S.E.M. All data were analyzed using GraphPad Prism version 7.0 software (San Diego, CA, USA). Differences between two groups were analyzed using unpaired two-tailed Students’ T-test. Two-way ANOVA and Bonferroni post-hoc analyses were used to calculate statistical differences between multiple groups. For all statistical tests, *p* < 0.05 was considered statistically significant.

## 5. Conclusions

The results showed that EEDAR can effectively alleviate diarrhea caused by spleen deficiency and improve intestinal pathological changes. Additionally, EEDAR can also inhibit decreased levels of tight junction protein and protect the integrity of the intestinal barrier through the P38 MAPK signaling pathway. Moreover, EEDAR can also regulate intestinal flora and improve its structure. These results assist in elucidation of the efficacy and mechanism of EEDAR and provide an important basis for rational clinical drug use.

## Figures and Tables

**Figure 1 ijms-21-00124-f001:**
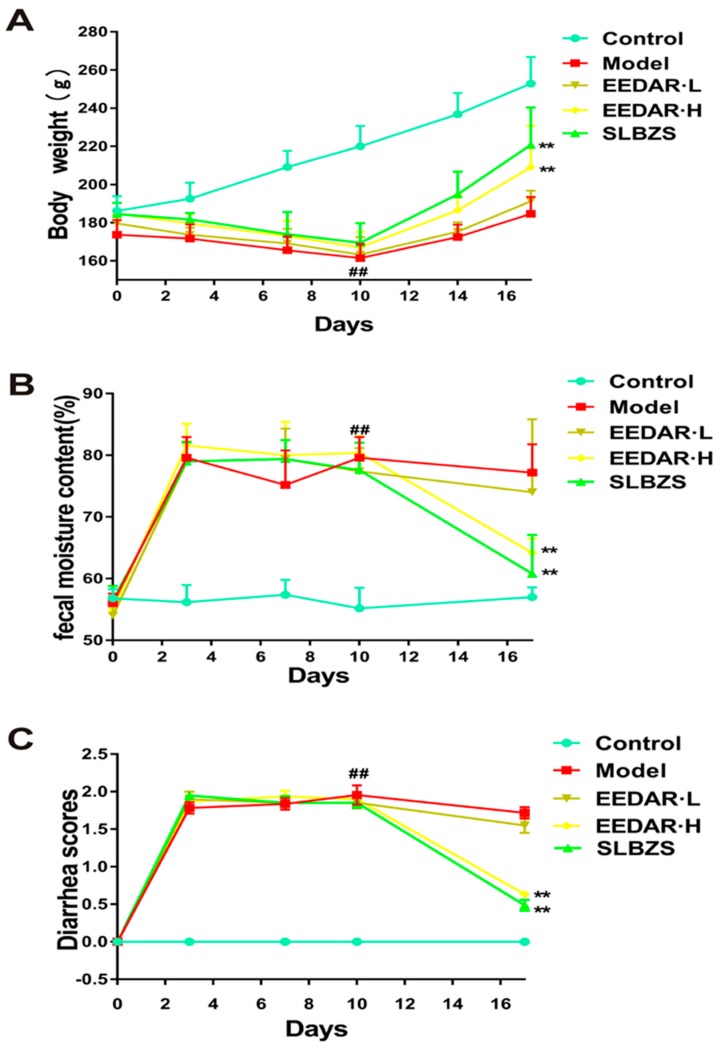
Effect of EEDAR on body weight, diarrhea scores, and fecal moisture content in spleen deficiency diarrhea rats (*n* = 10). Body weight (**A**), diarrhea scores (**B**), and fecal moisture content (**C**) changed daily during the experiment. Values are expressed as means ± SEM. ^##^
*p* < 0.01 vs. Control; ** *p* < 0.01 vs. Model.

**Figure 2 ijms-21-00124-f002:**
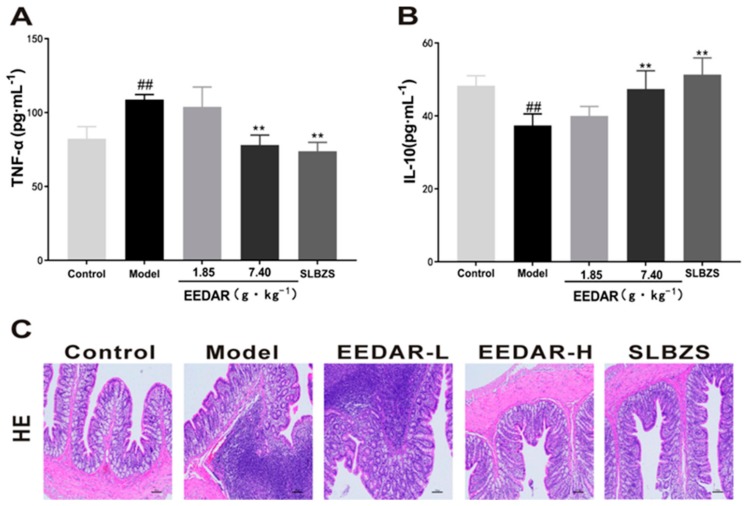
Effect of EEDAR on inflammation and the colons of spleen deficiency diarrhea rats (*n* = 10). The levels of TNF-α (**A**) and IL-10 (**B**) in serum were determined by ELISA. Representative images of the HE-stained sections (200× magnification) from the colons (**C**) indicated different pathological changes in spleen deficiency diarrhea rats when treated with EEDAR-L (1.85 g·kg^−1^), EEDAR-H (7.40 g·kg^−1^) and SLBZS. Values represent means ± SEM. ^##^
*p* < 0.01 vs. Control; ** *p* < 0.01 vs. Model.

**Figure 3 ijms-21-00124-f003:**
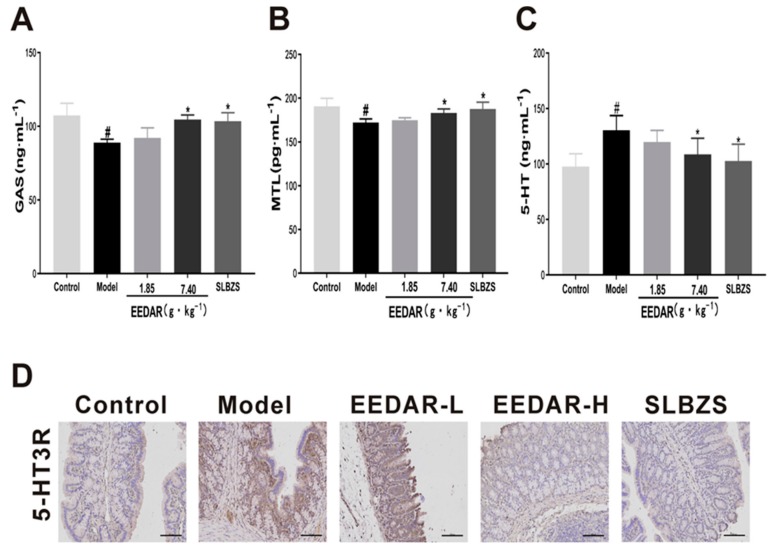
Effect of EEDAR on gastrointestinal motility hormone, serotonin, and its receptor in spleen-deficiency diarrhea rats (*n* = 10). Levels of GAS (**A**), MTL (**B**), and 5-HT (**C**) in serum were measured by ELISA. Representative images of 5-HTR3 (**D**) and IHC sections (200× magnification) from colons indicating different pathological changes in spleen deficiency diarrhea rats when treated with EEDAR-L (1.85 g·kg^−1^), EEDAR-H (7.40 g·kg^−1^), and SLBZS. Values represent means ± SEM. ^#^
*p* < 0.05 vs. Control; * *p* < 0.05 vs. Model. (Scale bar 1:100).

**Figure 4 ijms-21-00124-f004:**
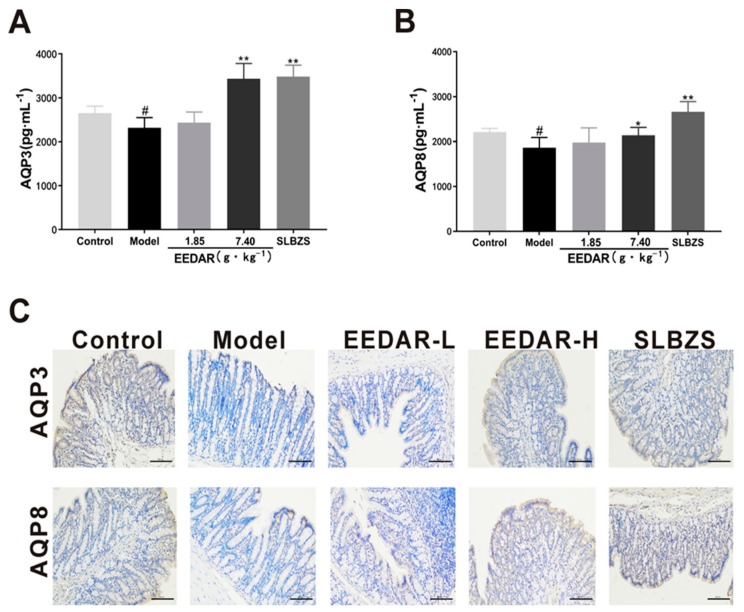
Effect of EEDAR on AQP3 and AQP8 in spleen-deficiency diarrhea rats (*n* = 10). The levels of AQP3 (**A**) and AQP8 (**B**) in the colon were determined by ELISA. Representative images of AQP3 and AQP8 (**C**) IHC sections (200× magnification) from colons indicated different pathological changes in spleen deficiency diarrhea rats treated with EEDAR-L (1.85 g·kg^−1^), EEDAR-H (7.40 g·kg^−1^), and SLBZS. Values represent means ± SEM. ^#^
*p* < 0.05, vs. Control; * *p* <0.05, ** *p* <0.01 vs. Model. (Scale bar 1:100).

**Figure 5 ijms-21-00124-f005:**
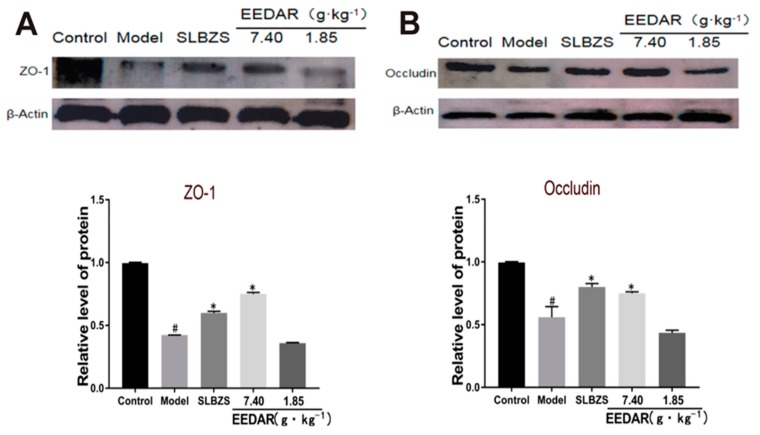
Effect of EEDAR on TJ protein expression in spleen-deficiency diarrhea rats (*n* = 3). ZO-1 (**A**) and Occludin (**B**) protein expression levels in colon tissue were measured by Western blot analysis. Expression bands were quantified, and values normalized to β-actin levels. Values represent means ± SEM. ^#^
*p* < 0.05 vs. Control; * *p* < 0.05 vs. Model.

**Figure 6 ijms-21-00124-f006:**
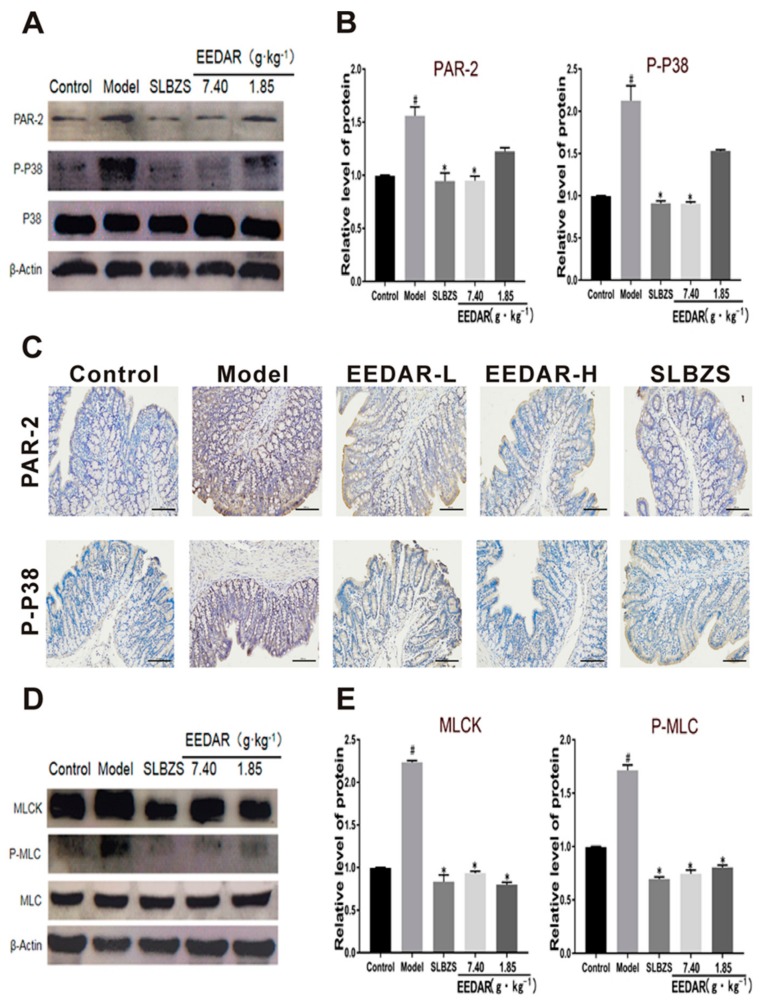
Effects of EEDAR on signaling molecules that regulate TJ structure and function in spleen-deficiency diarrhea rats (*n* = 3). The phosphorylation levels of p38 and PAR-2 in total colon homogenates were measured by Western blot analysis (**A**,**B**). Representative images of immunohistochemistry sections (200× magnification) from PAR-2 and P-P38 (**C**) colons indicated different pathological changes in spleen deficiency diarrhea rats treated with EEDAR-L (1.85 g·kg^−1^), EEDAR-H (7.40 g·kg^−1^), and SLBZS. Total protein levels of P-MLC and MLCK were measured by Western blot analysis (**E**,**F**). Bands were quantified and values normalized to non-phosphorylated protein or β-actin levels. Values represent means ± SEM. ^#^
*p* < 0.05 vs. Control; * *p* < 0.05 vs. Model. (Scale bar 1:100).

**Figure 7 ijms-21-00124-f007:**
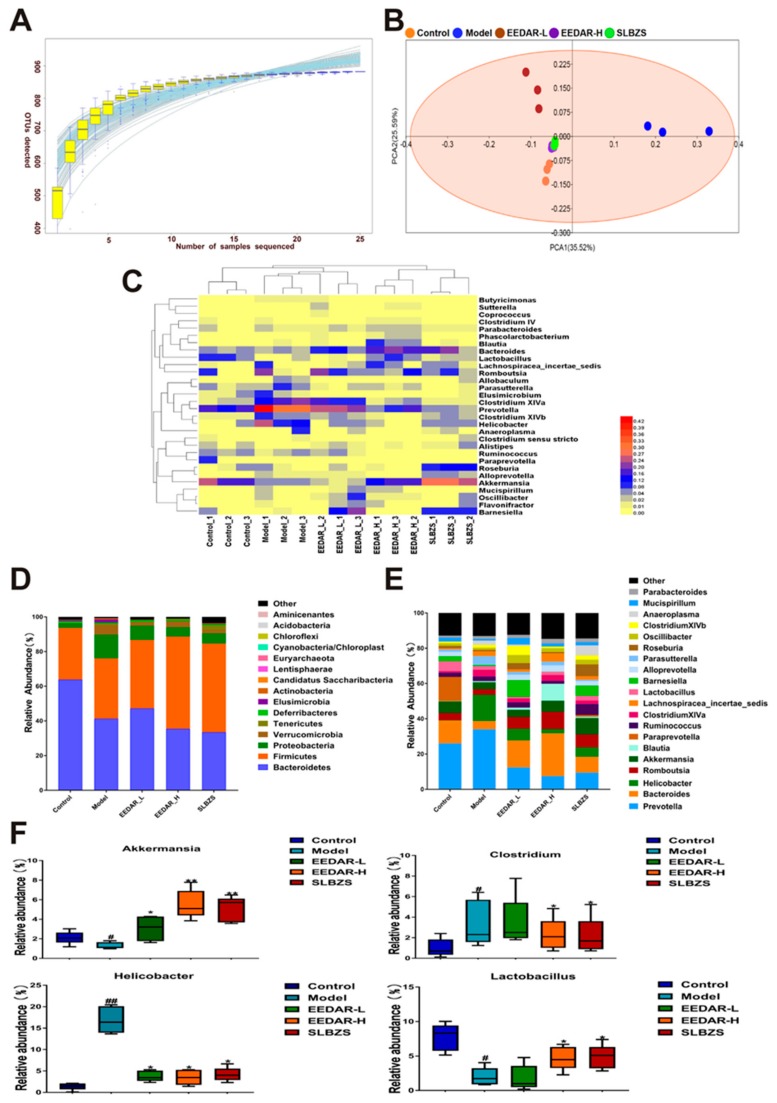
Effects of EEDAR on the overall structure of the gut microbiota in spleen-deficiency diarrhea rats (*n* = 3). (**A**) Sample number and species richness were estimated from a species accumulation boxplot. (**B**) PCA demonstrated distinct structural changes in the overall bacterial community for each group. (**C**) Heatmap demonstrating the relative abundance of intestinal bacteria at the genus level. (**D**) Microbial community bar plot at the phylum level. (**E**) Microbial community bar plot at the genus level. (**F**) Relative abundance of different microbial flora. Values represent means ± SEM. ^#^
*p* < 0.05, ^##^
*p* < 0.01 vs. Control; * *p* < 0.05, ** *p* < 0.01 vs. Model.

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
