# Peer review of "Deep-Fried Atractylodis Rhizoma Protects against Spleen Deficiency-Induced Diarrhea through Regulating Intestinal Inflammatory Response and Gut Microbiota"

_ijms, 2019, doi:10.3390/ijms21010124_

Round 1

Reviewer 1 Report

Thank you very much for opportunity to review this manuscript. Authors investigated the of EEDAR-H in the context of the spleen deficiency diarrhea syndrome.

However, I have few general remarks that authors should address properly.

Abstract is informative and well organized, however authors should highlight which changes are significant (it was used only once).

Introduction:

The role of microbiota is currently highly investigated in different diseases and most of the findings still “suggest” possible mechanism of action, which means that at least paragraph 48-62 should be carefully re-phrased. In general whole manuscript should be carefully checked whether authors express in English exactly this what they want-although manuscript is well-written. Readers do not need to be familiar with the name “Shen-Ling-Bai-Zhu-Scan” and the content should be introduce in the introduction section. Moreover, the is no information why investigating EEDAR-H in comparison to Shen-Ling-Bai-Zhu-Scan is valuable as this is another supplement…

Methods:

Line 328: I do have basic problem, which has to be clarify for Editor as well. Do authors really perform 16Sr DNA gene sequencing???? Or 16SrRNA???

Why authors did not compare EEDAR-H with ex. the routine used drug in diarrhea….? What authors obviously compare - the Control or “model group” should be described.

Authors should also add consistently the name of company, city and country to all reagents/kits that were used (relevant to all method section)

Results:

This section is properly described.

 Discussion

There is lack of citations so that could be that plagiarism will be detected and this part could be deeper discussed as authors obtained many results.

Author Response

Point 1: Abstract is informative and well organized, however authors should highlight which changes are significant (it was used only once).

Response 1: Thanks, we appreciate the comments. According to the reviewer’s comment, we have highlighted the changes are significant in the abstract in our revised manuscript. (See line 17--34)

Point 2: The role of microbiota is currently highly investigated in different diseases and most of the findings still “suggest” possible mechanism of action, which means that at least paragraph 48-62 should be carefully re-phrased.

Response 2: Thanks for your suggestions. According to the reviewer’s comment, we have added the detailed description about the possible mechanism, and cited recent papers in our revised manuscript. (See line 61-74)

Point 3: In general whole manuscript should be carefully checked whether authors express in English exactly this what they want-although manuscript is well-written.

Response 3: Thank you very much for your positive comments, We have been looking for a professional paper retouching company to provide language modification services for our papers and we have revised the paper carefully . (See revised manuscript)

Point 4:Readers do not need to be familiar with the name “Shen-Ling-Bai-Zhu-Scan” and the content should be introduce in the introduction section. Moreover, the is no information why investigating EEDAR-H in comparison to Shen-Ling-Bai-Zhu-Scan is valuable as this is another supplement…

Response 4:Thanks for your suggestions again. According to the reviewer’s comment, we have added the detailed description about “Shen-Ling-Bai-Zhu-Scan”, and cited recent papers in our revised manuscript. In China, Shen-Ling-Bai-Zhu-San (SLBZS) is widely used to treat diarrhea caused by spleen deficiency,so we think that investigating EEDAR-H in comparison to Shen-Ling-Bai-Zhu-Scan maybe  valuable.(See line 46-49)

Point 5:Line 328: I do have basic problem, which has to be clarify for Editor as well. Do authors really perform 16Sr DNA gene sequencing???? Or 16SrRNA???

Response 5:Thanks, we appreciate the comments. 16Sr DNA gene sequencing is right.

Point 6:Why authors did not compare EEDAR-H with ex. the routine used drug in diarrhea….? What authors obviously compare - the Control or “model group” should be described.

Response 6:Thanks, we appreciate the comments. In traditional Chinese medicine, there are many causes of diarrhea, and this paper mainly studies diarrhea caused by spleen deficiency. In China, Shen-Ling-Bai-Zhu-San (SLBZS) is often used to treat diarrhea caused by spleen deficiency, so this paper chooses Shen-Ling-Bai-Zhu-San (SLBZS) as a control. According to the reviewer’s comment, we have described the Control or “model group” in our revised manuscript.

Point 7:Authors should also add consistently the name of company, city and country to all reagents/kits that were used (relevant to all method section)

Response 7:Thanks for your suggestion, we agreed. According to the reviewer’s comment, we have added the reagents/kits Company, city, country and dilution ratio about each antibody in the part of method section in the revised manuscript.

Point 8:There is lack of citations so that could be that plagiarism will be detected and this part could be deeper discussed as authors obtained many results.

Response 8:Thanks for your suggestion, we agreed. According to the reviewer’s comment, we have cited more recent papers in our revised manuscript. Thanks for your suggestions again.

Reviewer 2 Report

This work is focus on the therapeutic effect and mechanism of DAR on the spleen deficiency diarrhea syndrome and provide a basis for the rational application of DAR.

Comments:

The use of mice or rats should be clarified. Auhors wrote mice in discussion line 279 (por instance) but rats in all results The treatment was with EEDAR-H and EEDAR-L and results obtained were different. In fact, EEDAR-H and SLBZS has similar behaviour in expression of GAS MTL and 5HT, diarrea reduced and fecal moisture content, but different after EEDAR-L. In discussion authors should be explain if they are talking about L or H. Figure 2. Add which data is L or H EEDAR product

Author Response

Point 1:The use of mice or rats should be clarified. Auhors wrote mice in discussion line 279 (por instance) but rats in all results.

Response 1:Thanks, we appreciate the comments. We have corrected the mice to rats and revised the paper carefully.

Point 2:The treatment was with EEDAR-H and EEDAR-L and results obtained were different. In fact, EEDAR-H and SLBZS has similar behaviour in expression of GAS MTL and 5HT, diarrea reduced and fecal moisture content, but different after EEDAR-L. In discussion authors should be explain if they are talking about L or H. Figure 2. Add which data is L or H EEDAR product

Response 2:Thanks for your suggestion, we agreed. According to the reviewer’s comment, we have clarified about EEDAR-H or EEDAR-L in discussion. (See line219-277). At the same time, we have supplemented and explained which data is L or H EEDAR product in the note of Figure 2.(See line116-117). Thanks for your suggestions again.

Round 2

Reviewer 1 Report

Authors addressed all major concerns, so I recommend to accept the manuscript for publication

Reviewer 2 Report

accepted